# Energy-Efficient CuO/TiO₂@GCN Cellulose Acetate-Based Membrane for Concurrent Filtration and Photodegradation of Ketoprofen in Drinking and Groundwater

**Lethula E. Mofokeng** [1,2]**, Lerato Hlekelele** [1,]*****, **John Moma** [2]**, Zikhona N. Tetana** [2,3,4] **and Vongani P. Chauke** [1,]*****

[1] Council for Scientific and Industrial Research, Meiring Naude Road, Brummeria, Pretoria 0184, South Africa; 446418@students.wits.ac.za

[2] Molecular Sciences Institute, School of Chemistry, University of the Witwatersrand, Private Bag X3, Johannesburg 2050, South Africa; John.Moma@wits.ac.za (J.M.); Zikhona.Tetana@wits.ac.za (Z.N.T.)

[3] DST-NRF Centre of Excellence in Strong Materials, University of the Witwatersrand, Private Bag X3, Johannesburg 2050, South Africa

[4] Microscopy and Microanalysis Unit, University of the Witwatersrand, Private Bag 3, Johannesburg 2050, South Africa

***** Correspondence: LHlekelele@csir.co.za (L.H.); VChauke@csir.co.za (V.P.C.)

**Abstract:** Photocatalytic membranes possessing both photocatalytic and solid-liquid separation capabilities were developed. These materials are based on ternary 1% CuO/TiO₂@GCN (1:9) embedded on cellulose acetate (CA) via the phase inversion method. The CA membranes containing 0.1, 0.3 and 0.5 wt% of 1% CuO/TiO₂@GCN (1:9) (CTG–100, CTG–300 and CTG–500) were fabricated. The deposition of 1% CuO/TiO₂@GCN (1:9) onto the CA membranes and the consequential changes in the materials' properties were investigated with various characterization techniques. For instance, PXRD, FTIR, and XPS analysis provided evidence that photocatalytic membranes were formed. Electron microscopy and EDX were then used to visualize the photocatalytic membranes and show that the photocatalyst (1% CuO/TiO₂@GCN (1:9)) was well dispersed onto the CA membrane. On the other hand, the properties of the photocatalytic membranes were scrutinized, where it was found that the membranes had a sponge-like morphology and that was significantly less hydrophilic compared to neat CA. The removal of KP in water using CTG–500 exhibited over 94% efficiency, while 38% for neat CA was achieved. Water permeability flux improved with increasing 1% CuO/TiO₂@GCN (1:9) and hydrophilicity of the membranes. The electrical energy consumption was calculated and determined to be significantly lower than that of the CA membrane. The CTG–500 membrane after every cycle showed self-cleaning ability after operation in drinking and groundwater.

**Keywords:** photocatalytic membrane; filtration; degradation; ground and drinking water; electrical energy consumption

---

## 1. Introduction

The polymeric membrane filtration technique is a widely practiced strategy for the removal of environmental contaminants from water. This is because it is an energy-efficient procedure that is effective even at low concentrations of water contaminants [1]. Moreover, a lot of the polymeric membranes known have high mechanical strength, good thermal and chemical stability, acid and alkaline resistance, and they are cheap to make [2]. However, the industrial application of cellulose acetate based membranes, in particular, is restricted because they are super-hydrophobic, and that membrane fouling occurs quickly. The consequences of their hydrophobicity and propensity to foul are that their separation efficiency is reduced and so is their water flux, resulting in high electrical energy consumption ($E_{EC}$) as high pressure is required. Additionally, this leads to high running expenses and the lifespan of the membranes is significantly reduced [3]. Attempts have been made to regenerate

membranes, but these processes usually involve the use of harsh chemicals and/or physical and electrical processes that can damage and contaminant the membrane [4].

The employment of microfiltration membrane (MF, 100–1000 nm pore sizes) is of interest recently for the eradication of micropollutants, whereas ultrafiltration (UF) (5–50 nm), nanofiltration (NF) (2–5 nm) [5], are mainly utilized to reject small solutes at high pressures (100–1000 psi) as compared to MF membranes [6]. MF membranes coupled with photocatalysis offer numerous advantages, such as simultaneous degradation and rejection of contaminants, preventing catalyst detachment, easy to recycle and reuse, controllable process relation to flow rate and filtration efficiency and compatibility with various reactors.

Perhaps the biggest problem with membrane technology for water purification is that the process generates sludge which is difficult and expensive to dispose of [4].

One of the strategies used to overcome setbacks associated with membrane technology is to integrate the membrane filtration process and with nanomaterials with photocatalytic properties. This process equips membrane materials with new properties, such as fouling resistance, high permeability, and photocatalytic activity [7–10]. Additionally, incorporating photocatalytic nanoparticles with membranes makes nanotechnology photocatalysis much more viable as it prevents the catalyst nanoparticles from leaching, and the nanoparticles become easier to recycle and reuse [2].

In the construction of photocatalytic membranes, the choice of the photocatalyst is crucial and $TiO_2$ has emerged to be one of the most popular semiconductors for this application [11–15]. This is not surprising as $TiO_2$ is the most widely used photocatalyst [16,17]. However, the large-scale application of $TiO_2$-nanoparticles as a photocatalyst is hampered because it is a wide bandgap semiconductor, the nanoparticles are difficult to separate from the treated water, and that its quantum efficiency (which is directly proportional to charge separation) is low [17–19]. This work seeks to address these issues by integrating different ideas that have been shown to work into a novel, efficient and effective material.

Here, we developed a membrane consisting of $TiO_2$ nanoparticles doped with CuO embedded onto graphitic carbon nitride (GCN) and cast onto cellulose acetate (CA) membrane. The configuration of the membrane was 1% $CuO/TiO_2$ and immobilized on graphitic carbon nitride at a ratio of 1:9 (1% $CuO/TiO_2$:GCN). The purpose of doping $TiO_2$ with CuO was to reduce the bandgap of $TiO_2$ while immobilizing $CuO/TiO_2$ onto graphitic carbon was to enhance charge separation thus increasing the quantum efficiency of the material. CuO was selected because of its narrow bandgap that ranges between 1.2 and 1.7 eV thus its presence will allow for the absorption of visible light, thus increasing the amount of light absorbed [19]. Furthermore, CuO is a p-type semiconductor and is capable of acting as a trap for the photo-excited electrons when used in conjunction with $TiO_2$ [19]. On the other hand, GCN is a semiconductor with a bandgap of ca. 2.7 eV, because of its $sp^2$ hybridized carbon and nitrogen atoms arranged in a six-member ring [20–24]. This along with the excellent thermal and chemical stability with optoelectronic properties of GCN, this material has been used as a photocatalyst [20,22–24]. Unfortunately, the use of GCN is limited by low visible light absorption, the short lifespan of photo-generated electron-hole pairs and low surface area [22]. One of the most viable ways of applying GCN in photocatalysis is compositing it to a wide bandgap photocatalyst which is possible because of its delocalized conjugated π structure [22].

Modifying the CA membranes was done to exploit synergetic properties between CA and CuO, $TiO_2$ and GCN sheets such that visible light absorption is achieved, fouling is suppressed, improve water flux and separation efficiency towards water samples spiked with ketoprofen as a model of contaminant. Inducing the self-cleaning and reusability of the modified CA membranes through visible light irradiation to enable $CuO/TiO_2$@GCN on the surface of the membrane to absorb and generate reactive radicals and degrade contaminants before the next cycle. Synergy is vital in advanced oxidation processes of all kinds [25–27]. These materials would ideally be incorporated into existing wastewater treatment processes pre-chlorination to limit the use of toxic chlorine.

## 2. Experimental

### 2.1. Materials

The chemicals and reagents used in this work were of analytical grade and were used without any further purification. Copper sulfate, nitric acid, hydrochloric acid, sodium hydroxide, tri-potassium citrate dihydrate, melamine, titanium (IV) isopropoxide (TTIP), ethanol, 2-propanol, and ketoprofen (KP) were all purchased from Sigma Aldrich. Ultrapure water was used throughout the experiments.

### 2.2. Preparation of $TiO_2$, $CuO/TiO_2$ Nanoparticles and Graphitic Carbon Nitride (GCN)

Synthesis of $TiO_2$ was prepared using the sol-gel method reported in the literature with minor modifications [28–31]. The typical method followed is described in the Supplementary Information document marked S1.

In the case of $CuO/TiO_2$ nanoparticles, the wet impregnation method was used. The details of these experiments are discussed in the Supplementary Information document marked S2.

The graphitic carbon was synthesized by polymerizing melamine as was described by synthesis of graphitic carbon nitride was prepared via thermal polymerization of melamine reported previously in the literature [32,33]. The full details are reported in the Supplementary Information marked S3.

### 2.3. Preparation of 1% $CuO/TiO_2$@GCN Photocatalysts

Preparation of GCN decorated with 1% $CuO/TiO_2$ was prepared via a simple sonication-assisted impregnation method [34]. The nanocomposites of 1% $CuO/TiO_2$ decorated on graphitic carbon nitride were dispersed in ethanol and sonicated for 30 min. Thereafter, the mixtures were magnetically stirred at 70 °C for 2 h. Thereafter, the obtained nanocomposites were centrifuged, washed several times with deionized water and ethanol. Finally, the nanocomposites were dried at 80 °C in a vacuum oven for 12 h and calcined in a muffle furnace at 450 °C for 2 h in the air at the rate of 10 °C/min.

### 2.4. Preparation of CA Membranes

The preparation of the CA membrane was achieved by the phase-inversion method adapted from the previous study with minor modifications [35,36]. The full details of this part of our work are described in the Supplementary Information document, labeled S4.

### 2.5. Preparation of $CuO/TiO_2$@GCN-CA Photocatalytic Membranes

The fabrication of cellulose acetate (CA) photocatalytic membranes was prepared through the phase-inversion method. The procedure is represented graphically in Supplementary Section S5, Figure S1. Typically, desired amounts of 1% $CuO/TiO_2$@GCN (1:9) (0.1, 0.3 and 0.5 wt%) were dispersed in DMSO (85 mL) using an ultrasonicator for 1 h. Thereafter, a constant amount of CA (15 g) was added into the mixture and dissolved at 60 °C for 6 h. The polymeric mixture was degassed for 12 h by bubbling with $N_2$ gas. A clean glass plate was used to cast the polymeric mixture using a casting knife. After casting, the glass plate was immediately immersed in a coagulation bath filled with deionized water to remove DMSO residues and the formation of membrane film occurred. Then, the wet membranes were transferred into fresh deionized water for 12 h to remove impurities and dried at room temperature. The compositions of the photocatalytic membranes are listed in Table 1. This process was selected as it prevents the NPs from leaching after the phase inversion fabrication method, as compared to the grafting polymerization method used on RO membranes [37,38].

**Table 1.** Chemical composition of casting solution for fabrication of $CuO/TiO_2@GCN$.

| Membrane ID | CA:DMSO Ratio | PVP/wt% | 1% $CuO/TiO_2@GCN$ (9:1) Content/wt% |
|:---:|:---:|:---:|:---:|
| **CA** | 15:83 | 2.00 | 0.00 |
| **CTG–100** | 15:83 | 2.00 | 0.10 |
| **CTG–300** | 15:83 | 2.00 | 0.30 |
| **CTG–500** | 15:83 | 2.00 | 0.50 |

### 2.6. Materials Characterization

The powder X-ray diffraction (PXRD) analysis of the photocatalysts and the membranes was conducted on a Bruker D2, 30 kV, 10 mA utilizing monochromatic CuKα radiation (k = 1.54184 Å) from 5–90°. Fourier transform infrared resonance (FTIR) analyses were conducted on a Perkin Elmer spectrum 100 spectrometer. The surface and morphological properties including elemental mapping were analyzed using scanning electron microscopy (SEM), FEI Quanta 400F equipped with energy-dispersive X-ray spectroscopy (EDS). Transmission electron microscopy (TEM, FEI Tecnai T12 operating at 200 kV), was also utilized to evaluate the morphological properties of the materials. The absorption spectra of the photocatalysts were evaluated using an Ultraviolet-visible spectrophotometer (Mettler Toledo, UV5Bio spectrophotometer, Columbus, OH, USA). The Kubelka-Munk method was utilized to determine the bandgap of the catalysts.

The hydrophobicity of the as-prepared membranes was analyzed using the sessile drop method operated on a drop shape analyzer (DSA 100, Krüss scientific, Hamburg, Germany). The measurements were conducted in triplicates at every 20 s interval. The thickness of the membranes was measured with a Vernier caliper (Absolute Digital Caliper, Johannesburg, South Africa). Thermogravimetric analysis (TGA) of the membranes was conducted using a thermogravimetric analyzer (NETZSCH, TG 209F1, Selb, Germany).

The membranes were cut into desired sizes and heated from 30 to 850 °C at a heating rate of 10 °C/min in the presence of a nitrogen atmosphere at a gas flow of 20 mL/min. X-ray photoelectron spectroscopy (XPS) operated at X-ray power of 300 W was utilized to evaluate the oxidation state and elemental composition of the membranes.

### 2.7. Flux and Antifouling Properties

The permeation flux performance of $CuO/TiO_2@GCN$ (9:1)-CA membrane with an effective area of 17.35 cm$^2$ was evaluated using a bench-scale filtration system consisting of Buchner funnel and glass filter flask connected and a vacuum pump (KNF, Freiburg, Germany) with maximum pressure and power of 1.0 bar and 100 W, respectively. The configuration of our system is shown in S6, Figure S2 (without the light). The membranes were immersed in deionized water for a period of 24 h before water permeability investigation of pure water flux. Steady vacuum pressure of 1.0 bar was the driving force for retention of water filtrate. Water permeate was monitored and collected continuously.

### 2.8. Evaluation of the Efficiency of the Photocatalytic Membrane

The efficiency at which the photocatalytic membranes were able to remove ketoprofen was assessed using a setup shown in S6, Figure S2. The lamp (165 W, Radiant, λ > 400 nm with an intensity of 240 W/m$^2$) was placed such that the tip of the lamp was 10 cm above the reactor vessel. Before passing through the contaminated feed, deionized water was passed through the membranes put in place in the setup for 30 min. The temperature surrounding the filtration system was maintained at 25 ± 2 °C throughout all the experiments. The experiments were conducted using 100 mL test solutions of 10 ppm ketoprofen in deionized, drinking, or groundwater.

These water samples were added to the Buchner funnel before vacuum filtration. During filtration, at every 1 h period, 2 mL filtrates of the test solution were collected from the glass filter (S6, Figure S2) and analyzed using UV-Vis spectrophotometer (Mettler Toledo, UV5Bio spectrophotometer, Columbus, OH, USA) monitored at 260 nm.

Also, an LC/MS (Nexera LC-40) system coupled with a Quadrupole Time-of-Flight mass spec operated in a negative ESI mode was used to separate and detect the degradation by-products. Here, a volume of 10 µL of the water samples was injected into a Shim-pack Velox SP-C18 column (2.1 × 100 mm, 2.7 µm) at 40 °C. The constituents were separated using an isocratic mobile phase consisting of (0.1% *v/v*) formic acid in 45% HPLC grade acetonitrile and 55% Milli-Q water at a flow rate of 1 mL/min. The total organic content (TOC) measurements were undertaken using a TOC analyzer (Tekmar Dohrmann Apollo 9000, Mason, WA, USA) dependent on the concentration of $CO_2$ for quantification.

## 3. Results and Discussion

### 3.1. Material Characterizations

The photocatalytic membranes reported in this work are based on the loading photocatalyst (1% $CuO/TiO_2$@GCN (1:9)) onto the cellulose membrane. Before loading the photocatalyst onto the membrane and comparing its properties relative to those of the resultant photocatalytic membrane, the morphological and optical properties of 1% $CuO/TiO_2$@GCN (1:9) were studied. It was found that the morphology of the photocatalyst consisted of 1% $CuO/TiO_2$ nanoparticles distributed on the surface of GCN (S7, Figure S3). The photocatalyst was found to have a narrow bandgap of less than 1.8 eV (S8, Figure S4). A more detailed discussion of this part of the work is presented in Supplementary Sections S7 and S8 for morphological and optical properties, respectively.

### 3.2. PXRD Analysis

The crystallinity of the nanocomposites consisting of 1% $CuO/TiO_2$@GCN (1:9) nanocomposite and 1% $CuO/TiO_2$@GCN (1:9)-CA (CTG–500) membrane was investigated using PXRD. The patterns are shown in Figure 1. The CA membrane pattern was characterized by two prominent peaks 2θ positions of 8° and 18°. These peaks are attributable to the CA membrane as it is semicrystalline [39,40].

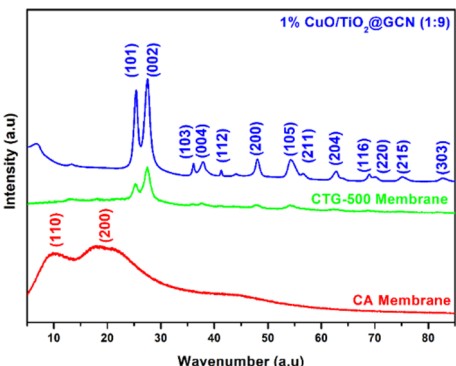

**Figure 1.** XRD patterns of 1% $CuO/TiO_2$@GCN (1:9) powder and 1% $CuO/TiO_2$@GCN (1:9)-CA (CTG–500) membrane and cellulose acetate (CA) membrane.

The characteristic peaks of anatase-$TiO_2$ in 1% $CuO/TiO_2$@GCN (1:9) were observed at 2θ values of 25.3°, 36.1°, 37.9°, 41.4°, 48.0°, 54.2°, 56.6°, 62.4°, 67.0°, 70.1°, 75.2°, and 82.7° were attributed to the diffraction planes (101), (103), (004), (112), (200), (105), (211), (204), (116), (220), (215) and (303), respectively [12,41]. The only peak attributable to GCN is the (002) reflection at 2θ = 27.6°, attributable to an extended range of layer by layer aromatic repeating units [42]. Peaks characteristic of CuO were not observed as it was used in small quantities (1%), relative to the very crystalline $TiO_2$.

The PXRD peaks attributable to 1% $CuO/TiO_2$@GCN (1:9) nanocomposites upon its incorporation into the CA membrane were still observable with slight shifts in 2θ positions. This indicates that the photocatalyst was loaded onto the membrane matrix, with a possibility of interactions between the materials. The evidence suggests that the interaction that occurred between $CuO/TiO_2$@GCN and CA matrix were Van Der Waals

forces, i.e., hydrogen bonding formed between the hydroxyl group in the cellulose network and also the surface OH groups of the photocatalyst.

### 3.3. XPS Analysis

The elemental composition and chemical nature of the 1% CuO/TiO$_2$@GCN-CA membrane were evaluated using X-ray photoelectron spectroscopy (XPS). The combined survey spectra containing elemental compositions of Ti, Cu, O and N on the surface of 1% CuO/TiO$_2$@GCN-CA membrane are presented in Figure 2a. The XPS survey showed no notable evidence of impurities on a wide range spectrum of XPS (0–1200 eV)**.**

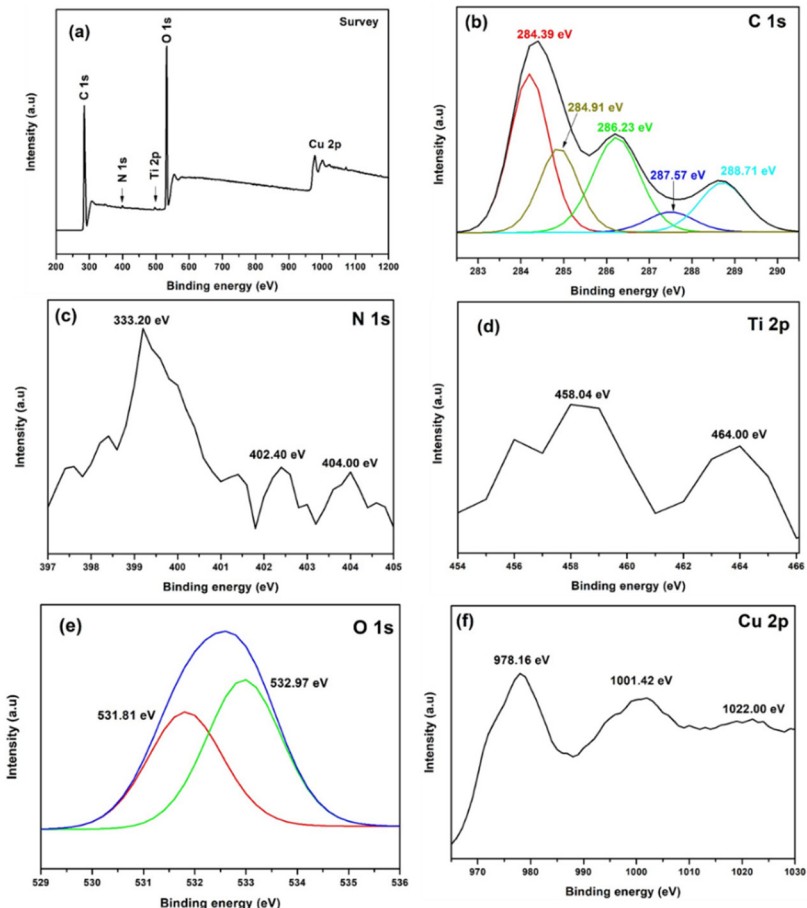

**Figure 2.** XPS spectra of 1% CuO/TiO$_2$@GCN-CA (CTG–500) membrane survey: (**a**) survey, (**b**) C 1s, (**c**) N 1s, (**d**) Ti 2p, (**e**) O 1s and (**f**) Cu 2p spectra.

Figure 2b showed five peaks of C 1s appearing at 284.39 and 284.91 eV, which were imputed to C-N and C-C on the GCN and CA, which suggested that the GCN incorporated with 1% CuO/TiO$_2$ within and on the surface of the CA membrane was achieved. The peak at 286.23 eV was attributed to C-O accredited to the methyl group bonded to an oxygen atom in CA. The peak at 287.57 eV was due to C=O functional group in CA, while the peak at 288.71 eV was due to carbon in the O-C-O functional group found in CA [43].

Figure 2c displayed three peaks at 333.20, 402.40, 404.00 eV indicating evidence of nitrogen N 1s emanating from the triazine structure. The peak appearing at 333.20 eV was imputed to the C-N=C sp$^2$-hybridized N atom found in triazine of GCN. The peak observed at 402.40 eV was attributed to bridged N-(C)$_3$. The peak at 404.00 eV was associated with the N atom in the amino group (N-H) [44].

Figure 2d showed two Ti characteristic peaks appearing at 458.04 and 464.00 eV which were assigned to Ti 2p$_{3/2}$ and Ti 2p$_{1/2}$, respectively emanated by Ti$^{4+}$ in TiO$_2$ [45]. This evinces that Ti$^{4+}$ ions in CuO/TiO$_2$ were incorporated successfully in the CA membrane.

Figure 2e displays O 1s spectrum consisting of two characteristic peaks appearing at 531.81 eV was ascribed to O-C which are found in CA and attributed to hydroxyl groups attached on the surface of the oxide, while the peak at 532.97 eV was ascribed to O=C which were imputed to $O^-$ ions in CuO or $TiO_2$ [43–46].

Figure 2f displayed Cu 2p spectrum with two key peaks at 978.16 and 1022 eV that were associated with the binding energies of Cu $2p_{1/2}$ and Cu $2p_{3/2}$, respectively which provided evidence of $Cu^{2+}$ in CuO [39]. The satellite peak appearing at 1001.42 eV is emanated by $Cu^{2+}$, which further confirms the presence of CuO in the CA membrane matrix [45].

### 3.4. FTIR Analysis

FTIR was used to identify the functional groups on the membrane, photocatalyst, and photocatalytic membrane. The spectra are shown in Figure 3. The spectrum of CA membrane was characterized mainly by bands at 2940 and 2870 $cm^{-1}$ attributable to the asymmetric and the symmetric C–H stretching, respectively. Additionally, the band at 1745 $cm^{-1}$ was attributed to the carbonyl group of the ester functional group while the band at 1168 $cm^{-1}$ was spoken for by the asymmetric stretching of the ether groups and the band at 1215 $cm^{-1}$ was associated with the carboxylate C–O stretch.

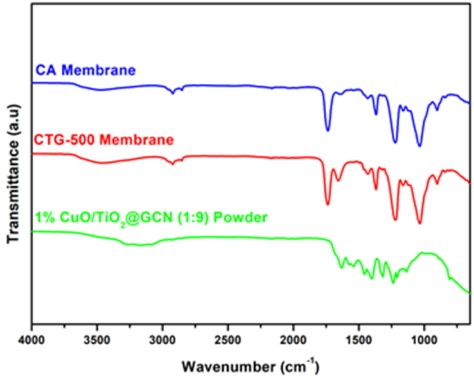

**Figure 3.** FTIR of CA membrane, CTG—500 membrane and 1% CuO/$TiO_2$@GCN (1:9) powder.

On the other hand, scrutinizing the FTIR spectrum of the photocatalyst, 1% CuO/$TiO_2$@GCN (1:9), evidence for $TiO_2$ nanoparticles and GCN was observed. For instance, the small band at 700 $cm^{-1}$ and the sharp decrease in transmission under 500 $cm^{-1}$ are attributable to the Ti−O−Ti stretching vibration. The presence of GCN was mainly signified by bands at 3100–3400 $cm^{-1}$ representing the N−H bond stretching vibration. The band at 1570 was associated with the C-N stretching while the bands at 1250, 1327, and 1420 $cm^{-1}$ were synonymous with the C-N stretching of the heterocyclic aromatic.

The FTIR spectrum of the photocatalytic membrane was observed to be a sum of the two other spectra with a few discrepancies. For instance, the band associable with Ti−O−Ti at 700 $cm^{-1}$ was not on the spectrum of the photocatalytic membrane. The ester band of the membrane at 1745 $cm^{-1}$ appeared to be more pronounced and at a shorter wavelength, relative to how it was observed on the spectrum of the membrane alone.

### 3.5. Membrane Characterizations

SEM Analysis of the Membranes

The surface and morphological properties of the pristine membrane CA and the photocatalytic membranes (ratios of the different membranes are given in Table 1) were investigated using SEM and the results are shown in Figure 4. Figure 4a shows a rough surface of CA consisting of pores that average 625 nm as shown in Figure 4b. These pores serve as passages for water to pass through and they are ideal for the immobilization of the photocatalyst particles [47]. The surface inspection of CTG–100, CTG–300, and CTG–500

was found to be similar to that of neat CA except that the photocatalytic membranes exhibited sponge-like structures with significantly diminished pore sizes. The average pore sizes of CTG–100, CTG–300, and CTG–500 were 381, 302 and 324 nm, respectively. It was observed that the pore sizes decreased with increasing concentration of 1% CuO/TiO$_2$@GCN (1:9) and thus, forming more sponge-like structures.

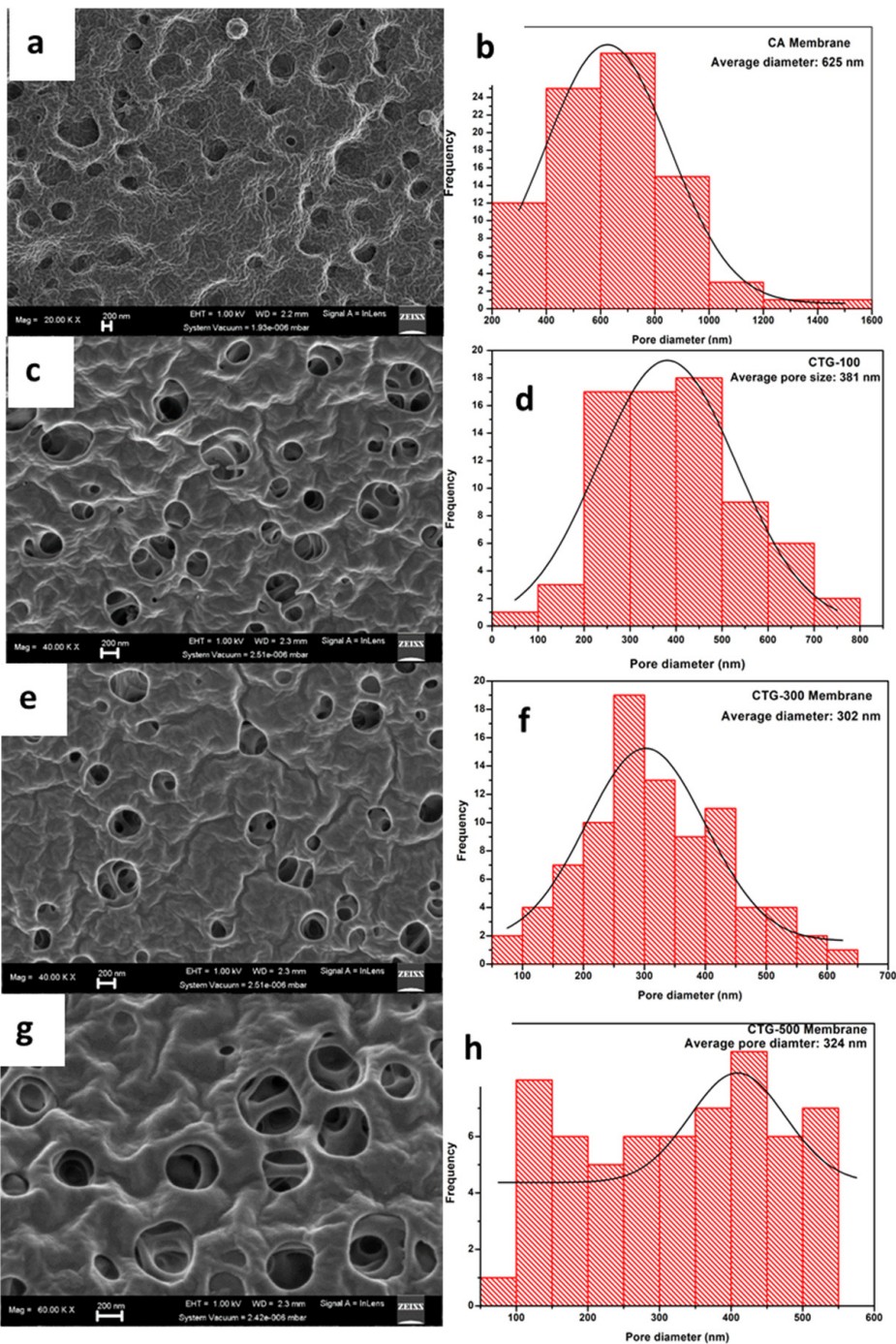

**Figure 4.** SEM micrographs of (**a**) neat CA, (**c**) CTG—100, (**e**) CTG—300 and (**g**) CTG—500 and their respective pore size distribution displayed in (**b**,**d**,**f**,**h**).

The formation of the sponge-like morphology was due to the strong affinity of 1% CuO/TiO$_2$@GCN (1:9) (0.5 wt%) to the CA membrane. This was a result of the rapid interchange between the solvent and non-solvent during the phase inversion fabrication

process [48]. The porous 3D interconnected structures extend within the membranes with almost visible circular pores for allowing water to pass through. Furthermore, this is beneficial for our cause as the 3D interconnected structure increases the absorption of light through the precise multireflection of the incoming rays [47].

### 3.6. Surface Hydrophilicity and Water Uptake Evaluation

Surface Hydrophilicity

The wettability evaluation of the membrane is an essential parameter to predict the interaction of the membrane films with water. The contact angle measurements were performed using the sessile drop method using water as a solvent for analyzing the hydrophobicity membranes. The results are shown in Figure 5 water droplets (20 µL) were placed onto the surface of the membrane for 20 s to obtain uniform dispersion.

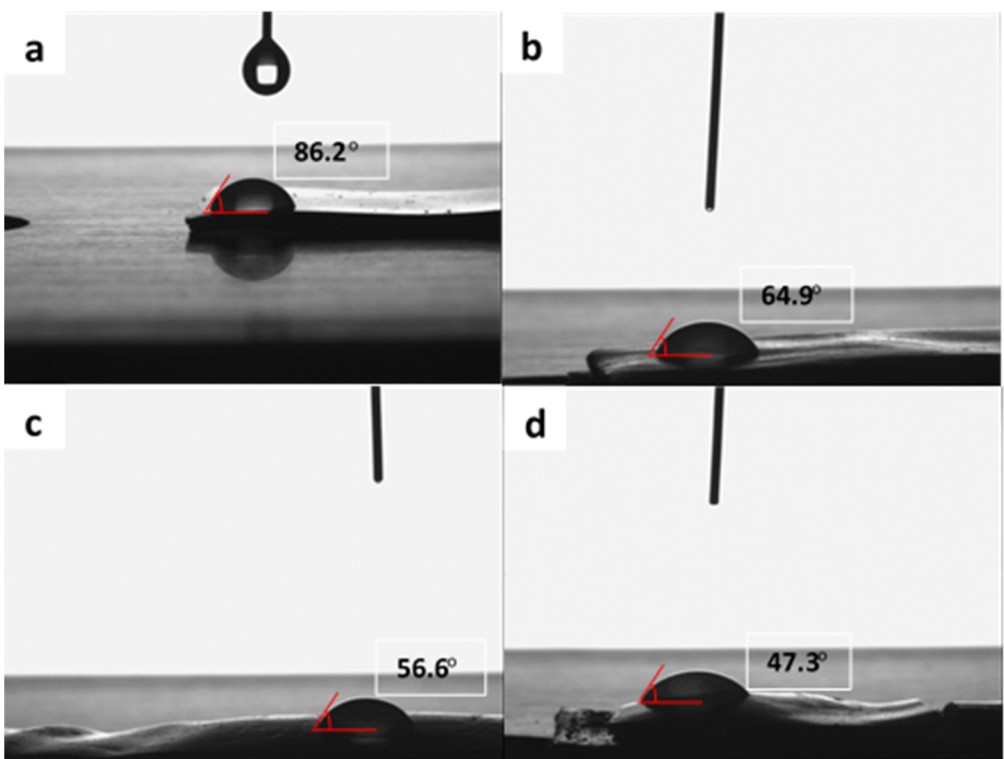

**Figure 5.** Contact angle micrographs of (**a**) CA, (**b**) CTG—100, (**c**) CTG—300 and (**d**) CTG—500 membranes recorded after every 20 s.

Figure 5a shows unmodified CA film exhibiting hydrophilic characteristics with an average contact angle of 86.2°. The modified CA films containing 0.1, 0.3, and 0.5 wt% $CuO/TiO_2@GCN$ (CTG–100, CTG–300, and CTG–500) as displayed in Figure 5b–d, showed improved hydrophilic properties with a reduced contact angle of 64.5, 56.6 and 47.3°, respectively. As the weight loading of $CuO/TiO_2@GCN$ was increased the hydrophilic nature of modified CA films increased. The co-existence of highly hydrophilic $CuO/TiO_2@GCN$ surface properties contributed to an increase in the hydrophilic properties of the modified CA films [3,49].

### 3.7. Water Uptake

The water uptake parameter of the membrane corresponds to the hydrophilicity characteristic of the membrane [40]. The water uptake of the membranes was calculated (Supplementary Section S9, Equation (S3)). The percentages of water uptake and porosities of CA and CTG–based membranes were calculated and presented in Table 2.

**Table 2.** Evaluation of membrane water uptake, porosity and shrinkage.

| Membrane ID | Water Uptake/% | Porosity/% | Shrinkage/% |
|---|---|---|---|
| CA | 72.00 | 79.42 | 31.85 |
| CTG–100 | 76.93 | 83.37 | 18.65 |
| CTG–300 | 78.06 | 84.25 | 17.68 |
| CTG–500 | 80.53 | 86.14 | 13.83 |

The water uptake for pristine CA was found to be 72.00% while those of the modified membranes, i.e., CTG–100, CTG–300, and CTG–500 membranes were 76.93, 78.06 and 80.53%, respectively. These results are consistent with the materials' affinity for water as it was shown in Figure 5, i.e., because CA is more hydrophobic than the modified membranes, it had the lowest water uptake. Additionally, the water uptake efficiency increased with the increasing concentration of the hydrophilic photocatalyst loaded onto the membrane.

*3.8. Permeability Evaluation*

Porosity and Shrinkage of the Membrane

Information and equations on how the shrinkage and porosity of the photocatalytic membranes were calculated are presented in Supplementary Section S9. The equation for shrinkage is Equation (S2) while that of porosity is Equation (S3). The results are tabulated in Table 2 and plotted in Figure 6. Here, it was observed that the porosity of the photocatalytic membranes was higher than that of pristine CA. Additionally, it was found that the porosity of the photocatalytic membranes increased with increasing amounts of the photocatalyst.

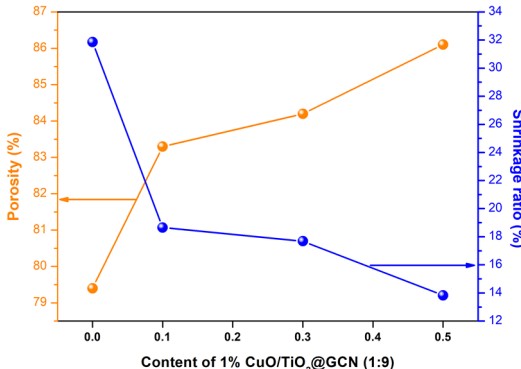

**Figure 6.** The plot of porosity and shrinkage ratio concerning the content of the dopant nanocomposite (1% CuO/TiO$_2$@GCN (1:9)).

In the case of the shrinkage ratio, it was observed that the photocatalytic membranes had a lower shrinkage ratio compared to the pristine CA membrane. Furthermore, it was observed that the shrinkage ratio decreased with the increasing concentration of the photocatalyst loaded onto the CA membrane.

These observations indicated that CuO/TiO$_2$@GCN (1:9) nanocomposite simultaneously improves the porosity, water uptake, shrinkage ratio (by decreasing it), and hydrophilicity [50].

*3.9. Pure Water Permeation and Membrane Recyclability Evaluation*

The pure water permeation and recyclability of the membranes are key aspects of the membranes for large-scale and real-life applications. The water permeation of the fabricated membranes per minute of filtration was assessed with and without visible light irradiation for five cycles. These measurements were done as it has been shown that the presence of light may result in the nanoparticles being affected in ways that may influence the membrane's permeability [1,51,52]. It is important to ascertain if changes that occur

to the photocatalytic membrane are due to the contaminant or experimental conditions, such as light. The membranes were flushed with water in between the cycles. The results are plotted in Figure 7. Here, it was found that the membranes showed good stability with minor reductions in pure water permeation when experiments were conducted in the presence of visible light. Additionally, it was observed that the CA membrane had a lower water permeation flux and filtration rate, which could be attributed to the hydrophobic nature of the membrane.

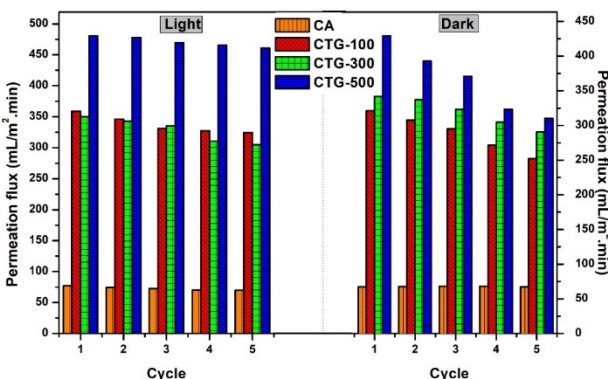

**Figure 7.** Recyclability of the as-prepared membrane and pure water permeability in the presence and absence of visible light.

Under visible light, the CA, CTG–100, CTG–300, and CTG–500 membranes exhibited average pure water permeability of 72.9, 329.6, 351.2 and 470.9 mL/m$^2$.min, respectively. It was noticeable that the water permeation flux increased slightly with the increasing concentration of 1% CuO/TiO$_2$@GCN (1:9) loading onto the CA matrix due to the increasing hydrophilicity of the membranes. This observation was consistent with the water contact angle measurements as shown in Figure 5. In the dark conditions and only flushing, the membrane with water before the next cycle, the CA, CTG–100, CTG–300, and CTG–500 membranes exhibited continuous declination in permeation flux with almost similar performance at the first cycle. The average water permeability was 67.7, 289.6, 319.6, and 365.3 mL/m$^2$.min for CA, CTG–100, CTG–300 and CTG–500, respectively.

In a study conducted by Zangeneh et al., it was reported that photocatalytic membranes containing TiO$_2$ exhibited elevated pure water fluxes under light irradiation as compared to dark surroundings [51]. Under light irradiation, the TiO$_2$ membranes absorb more water molecules due to the hydrophilic properties of the membranes [51,53,54]. This is because the radiation yields photoexcited electron-hole pairs, which are capable of reducing Ti$^{4+}$ to Ti$^{3+}$ and oxidizing O$_2^-$ anions through photoexcited holes. Then, the oxygen vacancies are produced on the membrane surface for the occupation of water molecules on these empty vacancies. Therefore, chemically bonded water molecules on the membrane surface can be adsorbed and passed through the membrane matrices continuously [51,53,54].

### 3.10. The Photocatalytic Performance of Modified CA Membranes

3.10.1. Photodegradation of KP in Deionized Water

The synthesized membranes, along with the pristine CA and the support (polyamide, PA) were used for the simultaneous filtration and degradation of KP in distilled water. Photolysis (without the membranes but in the presence of light) experiments were also conducted as a control. The photolysis experiment showed insignificant removal of KP (5.6%). Therefore, in experiments that involve the membranes, the reduction is attributable to the presence of the membranes (Figure 8 and Table 3). The reduction in the concentration of KP was calculated using Equation (S4) in Supplementary Document S10. The pristine membrane CA and the PA support exhibited low removal efficiencies, under 40% (Figure 8 and Table 3).

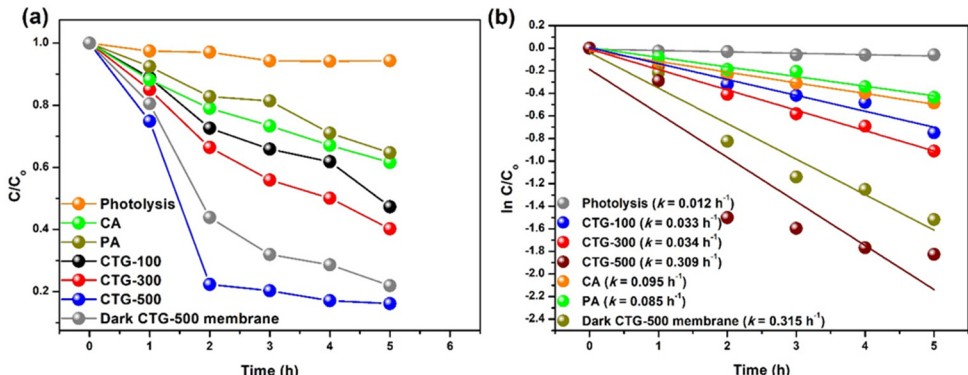

**Figure 8.** (**a**) Simultaneous filtration and photodegradation of KP (10 ppm, pH 5 for 5 h) using CTG—100, CTG—300, and CTG—500 membranes and (**b**) their reaction kinetics.

**Table 3.** The removal efficiency of the different membranes and electrical energy consumption associated with the process.

| Membrane ID | Rate Constant/$h^{-1}$ | Degradation Efficiency/% | $E_{EC}$ Costs/$m^3$ |
|---|---|---|---|
| CTG–100 | 0.141 | 46.1 | $5.92 \times 10^4$ |
| CTG–300 | 0.181 | 61.5 | $4.63 \times 10^4$ |
| CTG–500 | 0.390 | 83.7 | $2.18 \times 10^4$ |
| CTG–500 (in darkness) | 0.315 | 78.1 | $2.69 \times 10^4$ |
| CA | 0.095 | 38.3 | $8.75 \times 10^4$ |
| PA supporting membrane | 0.085 | 35.3 | $9.78 \times 10^4$ |
| Photolysis | 0.012 | 5.6 | $6.88 \times 10^5$ |

The photocatalytic membranes, i.e., samples CTG–100, CTG–300 and CTG–500, were found to have removal efficiencies higher than the pristine membranes. It was also observed that the removal efficiency of the photocatalytic membranes increased with an increase in the concentration of the photocatalyst (1% CuO/TiO$_2$@GCN (1:9)) loaded onto the CA membrane (Figure 8 and Table 3). The trend was justified because a decrease in the concentration of the photocatalyst loaded onto the CA membranes meant that the membranes were more hydrophilic (Figure 5) with smaller pores and inferior water uptake (Figure 6 and Table 2). Therefore, in the case of CTG–100, because of its low concentration of the photocatalyst, its pores were blocked quicker by the contaminant molecules thus reducing its filtration rate. This results in the formation of a thin layer of contaminant molecules forming on the surface of the photocatalytic membrane. This hinders the penetration of light, thus reducing the photocatalytic efficiency of the photocatalytic membranes.

Given that CTG–500 outperformed all the other membranes presented in this work, its removal efficiency without light was assessed. Here, the same experiment as above was conducted in the absence of light. It was found that its removal efficiency had decreased slightly from 83.9% to 78.1%.

The total organic content (TOC) which is a measure of the amount of organic compounds in the sample was measured to determine the amount of KP degraded in these experiments. Here, it was found that the TOC measured for the sample filtered in the dark was 42% higher than the TOC obtained when the same experiment was conducted in the presence of light. This indicates that 42% of KP was degraded instead of simply being adsorbed onto the membrane. The TOC of the original KP test solution was only 0.8% less than that of the experiments conducted in the dark, indicating that the experimental conditions in this work (other than light) did not have an effect on the degradation of KP. These results indicate that the impressive removal efficiency of CTG–500 is a result of a synergy between filtration and photocatalysis.

The cost associated with cleaning water is one of the factors that limit the wide-scale use of membranes and photocatalysis. The details of how the Electrical energy consumption ($E_{EC}$) was calculated are given in S11, Equation (S5). Here, it was observed that the high removal efficiency (also, the high rate constant) of CTG–500 translated into the process requiring the least amount of electrical energy to treat 1 L of 10 ppm KP than all the other materials tested in this work. Additionally, utilizing the CTG–500 membrane in the dark required lesser energy than all the other tested materials albeit in the dark. The electrical energy consumption followed a reverse trend to that of removal efficiency, i.e., in the order of decrease in consumption, CTG–500 > CTG–500 (in the dark) > CTG–300 > CTG–100 > CA > PA > photolysis. These results further signify the synergy between membrane technology and photocatalysis. Furthermore, it indicates that the increased hydrophobicity of the membranes positively affects the energy consumption of the process.

### 3.10.2. Photodegradation of KP in Drinking and Groundwater Using CTG–500 and Costs Associated with the Process

The photodegradation of KP (as a contaminant model) in different water qualities was evaluated by utilizing 1% $CuO/TiO_2$@GCN (9:1) (0.5 wt%) blended in the CA matrix to form CTG–500 membrane. The characteristics of the water matrixes are given in Supplementary Information S12, Table S1) This was done to investigate the applicability of the material into real water qualities to evaluate applicability in the real world. The results are plotted and shown in Figure 9 and Table 4 The photocatalytic membrane showed rapid removal of KP producing 94.0 and 94.8% efficiency in drinking and groundwater, respectively, higher than when it was tested in a distilled water matrix. This was attributed to the matrix of the drinking water and groundwater containing ions and organic matter (Table S1) that blocked the pores on the membrane preventing KP from flowing through, thus increasing the filtration of KP [55,56]. Furthermore, the ions in drinking and groundwater allowed for a higher residence time for KP to be photodegraded on the surface of the membrane [55,56]. Unfortunately, this caused the fouling of the membrane which resulted in a reduced flux which in turn reduced the volume of water treated per hour. This resulted in a significant increase in electrical energy consumption (Table 4).

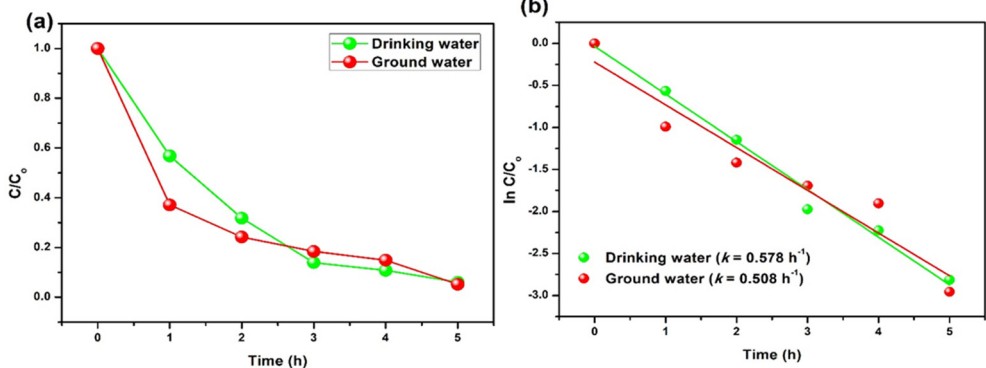

**Figure 9.** (**a**) Simultaneous filtration and photodegradation of KP (10 ppm, pH 5 for 5 h) in different water grades using CTG—500 membrane and (**b**) their reaction kinetics.

**Table 4.** Rate constants and energy consumption of the filtration process.

| Membrane ID | Rate Constant/h$^{-1}$ | Degradation Efficiency/% | $E_{EC}$/kWh/m$^3$ |
|---|---|---|---|
| CTG–500 (Groundwater) | 0.508 | 94.8 | $1.69 \times 10^4$ |
| CTG–500 (Drinking water) | 0.568 | 94.0 | $1.52 \times 10^4$ |

### 3.10.3. Photocatalytic Degradation of KP Drinking and Groundwater: Antifouling and Regeneration Studies in Drinking and Groundwater

The reusability and anti-fouling ability of the membranes are vastly necessary to assess their applicability in water treatment. The CTG–500 which exhibited better removal efficiencies was subjected to reusability studies for five cycles of the membrane filtration process using KP as a target contaminant in drinking and groundwater. To recycle the membrane, it was flushed with water in between the cycles and immersed in deionized water under the presence of light irradiation for 12 h. The results are shown in Figure 10.

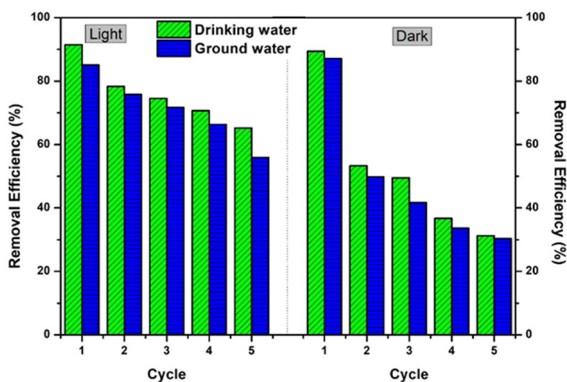

**Figure 10.** Removal of KP in drinking and groundwater using CTG–500 membrane.

The findings showed that the CTG–500 membrane possessed good reusability performance under visible light than in the dark. The removal of KP in drinking and groundwater in the first cycle was high, exhibiting 91.4 and 85.1% efficiencies for 5 h under visible light irradiation, respectively. On the other hand, when similar experiments were conducted in the dark, the removal of KP in drinking and groundwater in the first cycle was not significantly different from when the experiments were conducted in the presence of visible light. This meant that during the first cycle, the removal of KP was due to membrane filtration.

The removal efficiency gradually decreased and after the fifth cycle, it was 65.2% and 55.9% for drinking and groundwater, respectively (Figure 10). This was attributed to KP fouling on the surface of the membrane, thereby retarding light penetration towards reaching the photocatalytic membrane for the degradation process to occur. Nevertheless, the 26.2% loss in efficiency over five cycles was adjudged to signify that CTG–500 showed good self-cleaning capabilities. The self-cleaning capabilities of the membrane under visible light irradiation were investigated using FTIR (Figure S5, Supplementary Section S13). Here, no evidence of KP or other organic matter was observed on the surface of the membranes applied towards the removal of KP in the ground and drinking water.

However, the efficiency at which KP was removed in the ground and drinking water in the absence of light was greatly reduced. The first cycle exhibited 89.4% and 87.1% removal efficiencies for drinking and groundwater, respectively. The removal efficiencies decreased significantly after the second to the fifth cycle. These observations emanated from that, in the absence of light, the membrane only functioned as a filter thus it could not self-clean through photocatalysis. Therefore, the contaminant blocked the pores of the membrane by forming a thin layer on the surface of the membrane. This caused the membrane to foul thus reducing the filtration rate and permeate volume [52]. These results provide strong evidence that light irradiation on the photocatalytic membranes promotes antifouling capabilities and that the membrane could be re-used as a photocatalytic membrane.

### 3.11. Photocatalytic Mechanism
#### The Effect Scavengers

During the photocatalytic reaction, various reactive radicals, such as $H^+$, $HO^-$, $O_2^-$ and $e^-$ are produced which are responsible for the degradation of organic contaminants. Trapping of these reactive radicals using various scavengers was performed to evaluate how

the various reactive species influenced the removal of KP using the CTG–500 photocatalytic membrane under visible light irradiation. This was done in a batch study and the results are plotted in Figure 11 Ascorbic acid was used as an $H^+$ active species while isopropanol and $AgNO_3$ were used as scavengers for $HO^-$ and $e^-$ species, respectively. These scavengers were spiked into the KP test solutions.

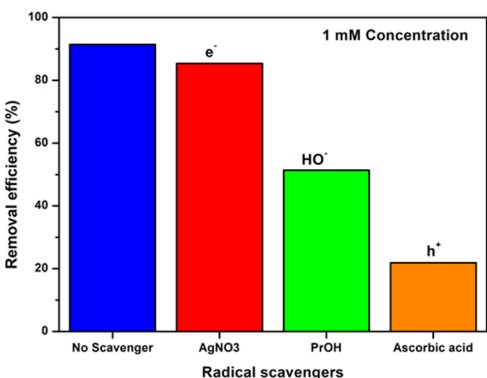

**Figure 11.** Photodegradation of KP in deionized water using CTG—500 membrane in the presence of various scavengers.

It was observed that in the absence of any scavenger, 91.4% removal efficiency of KP was achieved within 5 h under visible light irradiation. However, the removal efficiency KP decreased in the presence of $AgNO_3$, iso-propanol, and ascorbic acid yielding 85.4%, 51.3%, and 21.9%, respectively. These results indicated that electrons played an insignificant role in the degradation of KP while $HO^-$ ions played a moderate role and $H^+$ ions were the most active.

*3.12. Proposed Mechanism*

The removal of KP in distilled, ground, and drinking water using the CTG–500 photocatalytic membrane was found to have through filtration and photodegradation under light irradiation. During the filtration process, water molecules and KP contaminants are filtered through the pores of the membrane. The water molecules, because of their small diameter of 0.27 nm, were able to pass through the CTG–500 (with a pore size of 324 nm, Figure 4g,h) while the significantly larger KP was retained on the surface of the membrane. Additionally, the adsorption of KP onto the surface of CTG–500 was facilitated by hydrogen bonding, electrostatic interactions, and non-electrostatic forces that existed between the photocatalytic membrane and KP. In the case of non-electrostatic interactions, the attraction is between the GCN component of the photocatalytic membrane and free OH groups of KP. The structure of GCN, like that of nitrogen-doped carbon nanotubes and conductive polymers, such as polyaniline, consists of aromatic rings with delocalized π electrons; capable of forming π–π non-electrostatic interactions [25,57]. Hydrogen bonding was formed through the free hydroxyl groups on KP and the hydroxyl groups of CA [58,59].

Photodegradation was also key in the removal of KP, particularly for the self-cleaning characteristic of CTG–500 as it resulted in lower fouling and recyclability. This was possible because the $CuO/TiO_2@GCN$ (1:9) particles were successfully loaded onto the surface of CA to form the photocatalytic membrane. The photodegradation occurs via the absorption of visible light by the CuO dopant, yielding excitation of its valence band (VB) electrons. The bulk of the excited electrons then migrated into the conductive band (CB) through to the conductive GCN, which served as an electron sink. Some of the remaining electrons were able to react with the adsorbed $O_2$ molecules yielding superoxide radicals ($O_2\cdot$) which could react with KP. Nevertheless, as it was shown by studying the effect of scavengers, the electrons had a minimal role in the photodegradation of KP (Figure 11). On the other hand, the positively charged holes on the VB of CuO were left free to react with the adsorbed water

molecules to form hydroxyl radicals and hydronium ions. These species were the main actives responsible for the photodegradation of KP. The process is represented in Figure 12.

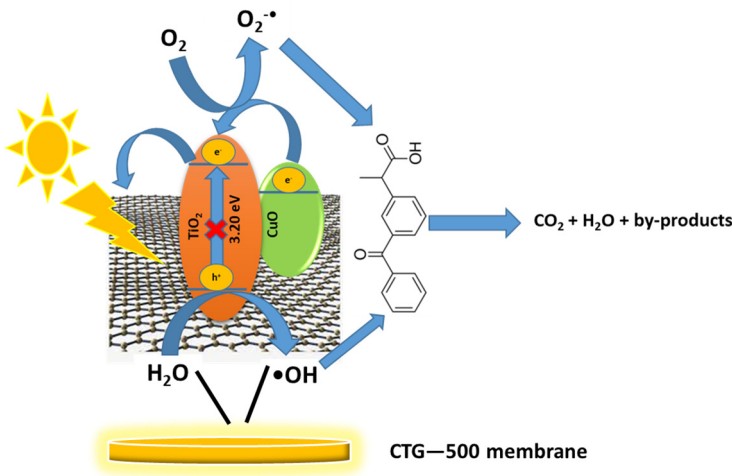

**Figure 12.** Proposed mechanism for photodegradation of KP using CTG—500 membrane.

*3.13. Proposed Degradation Pathway of KP*

The photodegradation and intermediates of KP were examined by LC-MS coupled with Q-TOF using the negative ion ESI mode. These experiments were conducted to further investigate if indeed KP was photodegraded in these experiments. The LC-MS data showed that the fresh KP test solution only had one KP peak while after 5 h of filtering in the presence of light, four additional peaks were observed within the measured mass range and elution time. The full description is given in S14. The chromatograms and mass spectrum are shown in Figure S6 while the proposed mechanism is in Figure S7.

## 4. Conclusions

In-situ fabrication of highly porous photocatalytic membranes based on ternary 1% CuO/TiO2@GCN (1:9) embedded in CA via phase inversion method was achieved successfully. The presence of ternary 1% $CuO/TiO_2$@GCN (1:9) nanocomposite within the membrane was confirmed by various characterizations techniques, such as XRD, FTIR, XPS and EDS. The absence of visible clustering of 1% $CuO/TiO_2$@GCN (1:9) nanocomposite within the CA membrane proved uniform dispersion was accomplished as observed on SEM micrographs (with 200 nm resolution).

The crystallinity of CA with an amorphous structure showed completely structural transition accredited to the embedding of 1% $CuO/TiO_2$@GCN (1:9) nanocomposite with exposed (101) and (002) peaks from $TiO_2$ and GCN, respectively, proving successful embedding of 1% $CuO/TiO_2$@GCN (1:9) within CA membrane.

The modified CA membranes containing different weight contents of 1% $CuO/TiO_2$@GCN (1:9) nanocomposites proved to elevate hydrophilicity, porosity, and the shrinkage nature of the membranes and thus, reducing the energy consumption of the membrane filtration system.

The modified CA membrane showed the ability to self-clean with the aid of 1% $CuO/TiO_2$@GCN (1:9) photocatalyst absorbing visible light and generating reactive radicals, thereby degrading traces of contaminants attached to the surface.

**Supplementary Materials:** The following supporting information can be downloaded at: https://www.mdpi.com/article/10.3390/app12031649/s1, S1: Method description: Preparation of $TiO_2$ nanoparticles. S2: Method description: Preparation of $CuO/TiO_2$ nanoparticles. S3: Method description: Preparation of $CuO/TiO_2$@GCN nanoparticles. S4: Method description: Preparation of cellulose acetate membranes. Figure S1: Systematic illustration for the fabrication of CA-based membranes embedded with ternary 1% $CuO/TiO_2$@GCN (1:9) nanocomposite. Figure S2: Filtration setup equipped with a medium power mercury lamp (165 W) and vacuum pump (100 W). S7: TEM

and SEM analysis of 1% CuO/TiO2@GCN nanocomposite. Figure S3: (a) TEM micrograph and (b) SEM of 1% CuO/TiO$_2$@GCN (1:9) nanocomposite, SEM elemental mapping of (c) Cu, (d) Ti, (e) C, (f) O and (g) EDX spectrum of 1% CuO/TiO$_2$@GCN (1:9). S8: Optical properties analyses of the various materials. Figure S4: UV/Vis spectra of (a) unmodified TiO$_2$ and 1% CuO/TiO$_2$@GCN (1:9) nano-composite and (b) their extrapolated energy bandgaps based on Kubelka-Munk method. S9: Method description of the analyses of membrane properties. S10: Method description of how the efficiency of the photocatalytic membrane was evaluated. S11: Description on how the electrical consumption was calculated. Table S1: The water quality characteristics of the different water matrixes used in this work. Figure S5: Series of steps for recycling the membranes and (d) ATR-FTR spectra of fresh CTG-500 membrane and after several cycles. Figure S6: Chromatograms of (a) uncatalyzed and (b) photodegraded KP after 5 h; (c) mass spectrum for a peak at retention time 3.09 min. Figure S7: Proposed photodegradation pathway and the intermediates of KP.

**Author Contributions:** Conceptualization, L.E.M.; methodology, L.E.M.; software, L.E.M. and L.H.; formal analysis, L.E.M., L.H. and V.P.C.; investigation, L.E.M.; validation, L.E.M., L.H. and V.P.C.; resources, V.P.C., Z.N.T. and J.M.; data curation, L.E.M., L.H. and V.P.C.; Writing Original Draft, L.E.M., L.H. and V.P.C.; writing—review and edit, L.H., V.P.C., Z.N.T. and J.M.; visualization, L.H., V.P.C., Z.N.T. and J.M.; supervision, L.H., V.P.C., Z.N.T. and J.M.; project administration, L.H., V.P.C., Z.N.T. and J.M.; funding acquisition, L.H., V.P.C., Z.N.T. and J.M. All authors have read and agreed to the published version of the manuscript.

**Funding:** This research was funded by National Research Foundation—Professional Development Programme (NRF-PDP) grant number (95564) under the Department of Science and Innovation, and the Council for Scientific and Industrial Research (CSIR) of South Africa (grant number 307775).

**Institutional Review Board Statement:** Not applicable.

**Informed Consent Statement:** Not applicable.

**Data Availability Statement:** The data presented in this study are available upon request from the corresponding author.

**Acknowledgments:** The authors would like to thank the University of the Witwatersrand and the Microscopy and Microanalysis Unit (MMU) for providing access to characterization facilities. The authors would like to appreciate the assistance of Abongile Jijana and Kabo Matshetshe on the characterization of materials at Mintek, South Africa.

**Conflicts of Interest:** The authors declare no conflict of interest.

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
