# Peer review of "Energy-Efficient CuO/TiO2@GCN Cellulose Acetate-Based Membrane for Concurrent Filtration and Photodegradation of Ketoprofen in Drinking and Groundwater"

_applsci, doi:10.3390/app12031649_

Round 1
Reviewer 1 Report
The content of the research is valuable and helpful for membrane preparation study and application in water treatment. Authors have carefully done a lot of experiments to explain their work. However, due to the complexity of the system investigated, including both a new photocatalyst and its related membrane, some more information are required to construct a clear illustration of the technology for readers. So i suggest authors add more data and improve the quality of the paper. Briefly:
- SEM image of NPs is required to know the size and morphology of the photocatalyst.
- Fig.7a, why the flux of CTG100 is higher than that of CTG300, explain it.
- Fig.10, the KP removal of CTG-500 is based on adsorption, so the adsorption amount of CTG-500 should be provided.
- ref. 56, please check it
- I can’t find the supplementary section of the paper.
Author Response
Point 1: SEM image of NPs is required to know the size and morphology of the photocatalyst.
Response: Thank you sincerely for giving us such thoughtful advice, which is very valuable in improving the quality of our manuscript. We have included a TEM and SEM images of TiO2 NP, 1% CuO/TiO2 nanocomposites along with its EDX spectrum. This is part of Figure S3 as part of S7. The following description was also added to S3.
Figure S3 are the (a) TEM and (b) SEM images of the photocatalyst, i.e. 1% CuO/TiO2. Here, it was observed that the nanocomposite consists of granular like nanoparticles that are agglomerated together. The size of the 1% CuO/TiO2 nanocomposites was determined by Sherrer’s equation and were found to be 18 nm. The EDX spectrum of 1% CuO/TiO2 nancomposite confirmed that the material consists Cu, O and Ti
Point 2: Fig.7a, why the flux of CTG100 is higher than that of CTG300, explain it.
Response: Thank you sincerely for your instructive advice. We repeated the measurements and realized that there was an error in reporting, we unfortunately had the values mixed up. The error is rectified on the main manuscript. The values are as follows; 72.9, 329.6, 351.2, and 470.9 mL/m2.min for CA, CTG-100, CTG-300, and CTG-500 membranes, respectively.
`
Point 3: Fig.10, the KP removal of CTG-500 is based on adsorption, so the adsorption amount of CTG-500 should be provided.
Response: Thank you for your valuable advice. The following description was added: On the other hand, when similar experiments were conducted in the dark, the removal of KP in drinking and groundwater in the first cycle was not significantly different to when the experiments were conducted in the presence of visible light. This meant that during the first cycle, the removal of KP was due to membrane filtration.
Point 4: ref. 56, please check it
Response: Thank you sincerely. The reference was corrected.
Point 5: I can’t find the supplementary section of the paper.
Response: Thank you for your valuable inputs. The supplementary data will be re-uploaded on the system.

Reviewer 2 Report
The manuscript is reporting the fabrication and the potential deployment of a new photocatalytic membrane. The authors assess various aspects of the technology, from the membrane fabrication procedure to its characterization, ending with a discussion related to the possible mechanisms behind the degradation of the contaminant. However, the manuscript needs a substantial revision prior to publication, due to lack of information and discussion mostly related to the usage of the membrane.
The abstract should be carefully revised, reporting a clear overview of the goal of the project, while focusing on the novelties reported in the manuscript. The hypothesis of the manuscript should be contextualized. The description of the membrane fabrication procedure can be shortened too.
Introduction:
It is confusing, since there is lack of a clear vision related to the membrane technology. Which membrane processes are covered? MF/UF or NF/RO, or even others? Please follow carefully the comments below.
Moreover, it’s important to define the application of the new membrane. There is no reason to fabricate a new membrane material without firstly having in mind its potential application, as well as its potential integration within a water/wastewater treatment plant. This is totally missed in the manuscript.
Line 61-66. Defining as “super hydrophobic” all the polymeric membranes is leading to misunderstanding, since classical TFC polyamide membrane for NF/RO applications are pretty hydrophilic instead…There is lack of references for this sentence too. Please revise the sentence.
Line 69-70. Membrane for “water purification” are generating sludge. Really? Can the authors make any example of it? Maybe brines, in the case of desalination systems. “Sludge” means an effluent full of organics, usually derived from biological treatment step. There is something confusing here.
Line 74-77. Most of the time the bottleneck of incorporating photocatalyst into membranes is exactly the leaching of the former one. Authors should better explain the concept reported.
Materials and methods.
It seems that a microfiltration membrane is developed in this work. However, it is not mentioned anywhere. Besides, there is a lack of characterization of the membrane, i.e., its permeability and selectivity, fundamental to define the type of membrane fabricated and its potential application.
Results and Discussion
Line 245-246. What does it mean “possibility of interaction”? Chemical or physical bonding? This is an important information in the view of the application, since physical bonding may lead to delamination of the membranes and the photocatalyst.
The authors should explain how the porosity and shrinkage would affect the performances of the membranes. This is missing in the manuscript.
Why did the authors perform the pure water permeability test with and without light? Pure water permeability should be an intrinsic characteristic of the membrane and shouldn’t be affected by the operating conditions (indeed, pure water permeability in pressure-drive membrane system is defined as LMH/bar). Moreover, the pure water permeability must be calculated measuring the pure water flux while varying the driving force (i.e., at different applied pressures). Here this is missed.
Line 443-444, “larger than”? something is missing
455-461. Authors should better explain how the filtration (a physical treatment process) would affect the degradation of the contaminant. I guess that part of the contaminant is instead removed, resulting in an increase in the concentration in the retentate. Did the authors measure the concentration of the contaminant in the retentate as well? There is lack of information to discuss the degradation of the contaminant by the membrane.
Table 3. Authors should be able to assess the degradation rates instead of the generic “removal efficiency”. This is something fundamental since one of the largest problem with membrane-based separation units is the disposal/management of the retentate produced. Photocatalytic membranes may be partially solve this issue while degrading some of the contaminants in solution.
The manuscript is pretty long. The proposed mechanism may be reported in the SI as well.
Author Response
Introduction
Point 1: It is confusing, since there is lack of a clear vision related to the membrane technology. Which membrane processes are covered? MF/UF or NF/RO, or even others? Please follow carefully the comments below.
Response: The following description was added to the introduction section: The employment of microfiltration membrane (MF, 100 – 1000 nm pore sizes) are of interest recently for the eradication of micropollutants, whereas UF (5 – 50 nm), NF (2 – 5 nm) [9], are mainly utilized to reject small solutes at high pressures (100 – 1000 psi) as compared to MF membranes [10]. MF membranes coupled with photocatalysis offer numerous advantages such as simultaneous degradation and rejection of contaminants, preventing catalyst detachment, easy to recycle and reuse, controllable process relation to flow rate and filtration efficiency and compatible with various reactors
Point 2: Moreover, it’s important to define the application of the new membrane. There is no reason to fabricate a new membrane material without firstly having in mind its potential application, as well as its potential integration within a water/wastewater treatment plant. This is totally missed in the manuscript.
Response: These materials would ideally be incorporated into existing wastewater treatment processes pre chlorination to limit the use of the toxic chlorine.
Point 3: Line 61-66. Defining as “super hydrophobic” all the polymeric membranes is leading to misunderstanding, since classical TFC polyamide membrane for NF/RO applications are pretty hydrophilic instead…There is lack of references for this sentence too. Please revise the sentence.
Response: Response: The sentence was qualified by being specific that cellulose acetate based membranes are super-hydrophobic.
Point 4: Line 74-77. Most of the time the bottleneck of incorporating photocatalyst into membranes is exactly the leaching of the former one. Authors should better explain the concept reported.
Response: Response: Thank you for your input. The following description was added to the methodology section: The process of embedding/blending the NPs with the CA matrix preserves the NPs from leaching after phase inversion fabrication method, as compared to grafting polymerization method used on RO membranes [7,15].
Moreover, the catalyst utilized exhibit optical properties under UV-Vis spectrophotometer. The absorbance of the leaching catalyst would have been detected when analysing the filtrate under UV-Vis spectrophotometer.
Materials and methods.
Point 5: It seems that a microfiltration membrane is developed in this work. However, it is not mentioned anywhere. Besides, there is a lack of characterization of the membrane, i.e., its permeability and selectivity, fundamental to define the type of membrane fabricated and its potential application.
Response: The following passages were added to the introduction section.
The employment of microfiltration membrane (MF, 100 – 1000 nm pore sizes) are of interest recently for the eradication of micropollutants, whereas UF (5 – 50 nm), NF (2 – 5 nm) [9], are mainly utilized to reject small solutes at high pressures (100 – 1000 psi) as compared to MF membranes [10]. MF membranes coupled with photocatalysis offer numerous advantages such as simultaneous degradation and rejection of contaminants, preventing catalyst detachment, easy to recycle and reuse, controllable process relation to flow rate and filtration efficiency and compatible with various reactors.
And
These materials would ideally be incorporated into existing wastewater treatment processes pre chlorination to limit the use of the toxic chlorine.
Results and Discussion
Point 1: Line 245-246. What does it mean “possibility of interaction”? Chemical or physical bonding? This is an important information in the view of the application, since physical bonding may lead to delamination of the membranes and the photocatalyst.
Response: The following description was added to the manuscript: The evidence suggests that the interactiona that occured between CuO/TiO2@GCN and CA matrix were van der Waals forces, i.e., hydrogen bonding formed between between the hydroxyl group in the cellulose network and also the surface OH groups of the photocatalyst.
Point 2: The authors should explain how the porosity and shrinkage would affect the performances of the membranes. This is missing in the manuscript.
Response: Thank you for your inputs. The calculations in which the membrane porosity and shrinkage ratio of the as-prepared membranes are given in the supplementary information, S9. The following description was also added: These measurements were conducted as the porosity and thickness of the membrane in the presence of water tend to increase membrane swelling. An increase in porosity of the membrane signifies that there is high volume of pores per total volume of the membrane [19], hence, the membranes with porosity (%) over 80% are desired [18,20]. Moreover, porous photocatalytic membranes tend to exhibit high porosity due to their excess pore structures. Shrinkage ratio is performed by measuring wet and dry membranes thickness and length. After drying membranes tend to shrink and thickness and length of the membrane is reduced. Reports suggested that lower shrinkage ratio below 20-25% are desired. The filtration process of high shrinkage ratio hinders water permeation flux and the rejection of the contaminant, by squeezing or blocking the pore passages [18].
Point 3: Why did the authors perform the pure water permeability test with and without light? Pure water permeability should be an intrinsic characteristic of the membrane and shouldn’t be affected by the operating conditions (indeed, pure water permeability in pressure-drive membrane system is defined as LMH/bar). Moreover, the pure water permeability must be calculated measuring the pure water flux while varying the driving force (i.e., at different applied pressures).
Response: Thank you for constructive comments. The following description was added to the manuscript. These measurements were done as it has been shown that the presence of light may result in the nanoparticles being affected in ways that may influence the membrane’s permeability [9–11]. It is important to ascertain if changes that occur to the photocatalytic membrane are due to the contaminant or experimental conditions such as light.
Point 4: Line 443-444, “larger than”? something is missing
Response: Thank you for your valuable inputs. The omitted word was added.
Point 5: 455-461. Authors should better explain how the filtration (a physical treatment process) would affect the degradation of the contaminant.
Response: The following description, which speaks to if the experimental conditions affected the degradation of KP is included in the manuscript: The total organic content (TOC) which is a measure of the amount of organic compounds in the sample was measured to determine the amount of KP degraded in these experiments. Here, it was found that the TOC measured for the sample filtered in the dark was 42 % higher than the TOC obtained when the same experiment was conducted in the presence of light. This indicates that 42 % of KP was degraded instead of simply being adsorbed onto the membrane. The TOC of the original KP test solution was only 0.8 % less than that of the experiments conducted in the dark, indicating that the experimental conditions in this work (other than light) did not have an effect on the degradation of KP.
Point 5: I guess that part of the contaminant is instead removed, resulting in an increase in the concentration in the retentate. Did the authors measure the concentration of the contaminant in the retentate as well? There is lack of information to discuss the degradation of the contaminant by the membrane.
Response: Thank you for your question. Unfortunately, only the concentration of the feed and permeate were measured. The measurements will be included in our future studies as this is at of ongoing work.
Point 6: Table 3. Authors should be able to assess the degradation rates instead of the generic “removal efficiency”. This is something fundamental since one of the largest problem with membrane-based separation units is the disposal/management of the retentate produced. Photocatalytic membranes may be partially solve this issue while degrading some of the contaminants in solution.
Response: Thank you sincerely for your instructive advice. The following description was added to the manuscript: The total organic content (TOC) which is a measure of the amount of organic compounds in the sample was measured to determine the amount of KP degraded in these experiments. Here, it was found that the TOC measured for the sample filtered in the dark was 42 % higher than the TOC obtained when the same experiment was conducted in the presence of light. This indicates that 42 % of KP was degraded instead of simply being adsorbed onto the membrane. The TOC of the original KP test solution was only 0.8 % less than that of the experiments conducted in the dark, indicating that the experimental conditions in this work (other than light) did not have an effect on the degradation of KP.
Point 7: The manuscript is pretty long. The proposed mechanism may be reported in the SI as well.
Response: Thank you for your valuable advice. The section that involves the proposed degradation pathway of KP were removed and inserted into SI.

Round 2
Reviewer 1 Report
it can be accepted now.
Reviewer 2 Report
The authors addressed the comment reported in the revision. For this reason the manuscript can be considered for publication in Applied Science.